# Convex Two-Layer Modeling

**Özlem Aslan      Hao Cheng      Dale Schuurmans**
Department of Computing Science, University of Alberta
Edmonton, AB T6G 2E8, Canada
{ozlem,hcheng2,dale}@cs.ualberta.ca

**Xinhua Zhang**
Machine Learning Research Group
National ICT Australia and ANU
xinhua.zhang@anu.edu.au

## Abstract

Latent variable prediction models, such as multi-layer networks, impose auxiliary latent variables between inputs and outputs to allow automatic inference of implicit features useful for prediction. Unfortunately, such models are difficult to train because inference over latent variables must be performed concurrently with parameter optimization—creating a highly non-convex problem. Instead of proposing another local training method, we develop a convex relaxation of hidden-layer conditional models that admits global training. Our approach extends current convex modeling approaches to handle two nested nonlinearities separated by a non-trivial adaptive latent layer. The resulting methods are able to acquire two-layer models that cannot be represented by any single-layer model over the same features, while improving training quality over local heuristics.

## 1   Introduction

Deep learning has recently been enjoying a resurgence [1, 2] due to the discovery that stage-wise pre-training can significantly improve the results of classical training methods [3–5]. The advantage of latent variable models is that they allow abstract "semantic" features of observed data to be represented, which can enhance the ability to capture predictive relationships between observed variables. In this way, latent variable models can greatly simplify the description of otherwise complex relationships between observed variates. For example, in unsupervised (i.e., "generative") settings, latent variable models have been used to express feature discovery problems such as dimensionality reduction [6], clustering [7], sparse coding [8], and independent components analysis [9]. More recently, such latent variable models have been used to discover abstract features of visual data invariant to low level transformations [1, 2, 4]. These learned representations not only facilitate understanding, they can enhance subsequent learning.

Our primary focus in this paper, however, is on conditional modeling. In a supervised (i.e. "conditional") setting, latent variable models are used to discover intervening feature representations that allow more accurate reconstruction of outputs from inputs. One advantage in the supervised case is that output information can be used to better identify relevant features to be inferred. However, latent variables also cause difficulty in this case because they impose nested nonlinearities between the input and output variables. Some important examples of conditional latent learning approaches include those that seek an intervening lower dimensional representation [10] latent clustering [11], sparse feature representation [8] or invariant latent representation [1, 3, 4, 12] between inputs and outputs. Despite their growing success, the difficulty of training a latent variable model remains clear: since the model parameters have to be trained concurrently with inference over latent variables, the convexity of the training problem is usually destroyed. Only highly restricted models can be trained to optimality, and current deep learning strategies provide no guarantees about solution quality. This remains true even when restricting attention to a single stage of stage-wise pre-training: simple models such as the two-layer auto-encoder or restricted Boltzmann machine (RBM) still pose intractable training problems, even within a single stage (in fact, simply computing the gradient of the RBM objective is currently believed to be intractable [13]).

Meanwhile, a growing body of research has investigated reformulations of latent variable learning that are able to yield tractable global training methods in special cases. Even though global training formulations are not a universally accepted goal of deep learning research [14], there are several useful methodologies that have been been applied successfully to other latent variable models: boosting strategies [15–17], semidefinite relaxations [18–20], matrix factorization [21–23], and moment based estimators (i.e. "spectral methods") [24, 25]. Unfortunately, none of these approaches has yet been able to accommodate a non-trivial hidden layer between an input and output layer while retaining the representational capacity of an auto-encoder or RBM (e.g. boosting strategies embed an intractable subproblem in these cases [15–17]). Some recent work has been able to capture restricted forms of latent structure in a conditional model—namely, a single latent cluster variable [18–20]—but this remains a rather limited approach.

In this paper we demonstrate that more general latent variable structures can be accommodated within a tractable convex framework. In particular, we show how two-layer latent conditional models with a single latent layer can be expressed equivalently in terms of a *latent feature kernel*. This reformulation allows a rich set of latent feature representations to be captured, while allowing useful convex relaxations in terms of a semidefinite optimization. Unlike [26], the latent kernel in this model is explicitly learned (nonparametrically). To cope with scaling issues we further develop an efficient algorithmic approach for the proposed relaxation. Importantly, the resulting method preserves sufficient problem structure to recover prediction models that cannot be represented by any one-layer architecture over the same input features, while improving the quality of local training.

## 2 Two-Layer Conditional Modeling

We address the problem of training a two-layer latent conditional model in the form of Figure 1; i.e., where there is a single layer of $h$ latent variables, $\phi$, between a layer of $n$ input variables, $\mathbf{x}$, and $m$ output variables, $\mathbf{y}$. The goal is to predict an output vector $\mathbf{y}$ given an input vector $\mathbf{x}$. Here, a prediction model consists of the composition of two nonlinear conditional models, $\mathbf{f}_1(W\mathbf{x}) \rightsquigarrow \phi$ and $\mathbf{f}_2(V\phi) \rightsquigarrow \hat{y}$, parameterized by the matrices $W \in \mathbb{R}^{h \times n}$ and $V \in \mathbb{R}^{m \times h}$. Once the parameters $W$ and $V$ have been specified, this architecture defines a point predictor that can determine $\hat{\mathbf{y}}$ from $\mathbf{x}$ by first computing an intermediate representation $\phi$.

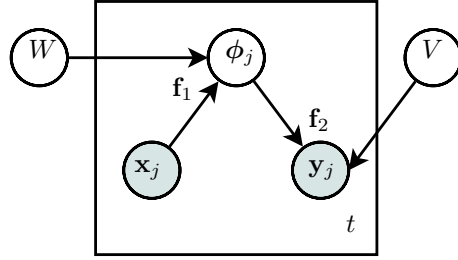

Figure 1: Latent conditional model $\mathbf{f}_1(W\mathbf{x}) \rightsquigarrow \phi, \mathbf{f}_2(V\phi) \rightsquigarrow \hat{y}$, where $\phi_j$ is a latent variable, $\mathbf{x}_j$ is an observed input vector, $\mathbf{y}_j$ is an observed output vector, $W$ are first layer parameters, and $V$ are second layer parameters.

To learn the model parameters, we assume we are given $t$ training pairs $\{(\mathbf{x}_j, \mathbf{y}_j)\}_{j=1}^{t}$, stacked in two matrices $X = (\mathbf{x}_1, ..., \mathbf{x}_t) \in \mathbb{R}^{n \times t}$ and $Y = (\mathbf{y}_1, ..., \mathbf{y}_t) \in \mathbb{R}^{m \times t}$, but the corresponding set of latent variable values $\Phi = (\phi_1, ..., \phi_t) \in \mathbb{R}^{h \times t}$ remains *unobserved*.

To formulate the training problem, we will consider two losses, $L_1$ and $L_2$, that relate the input to the latent layer, and the latent to the output layer respectively. For example, one can think of losses as negative log-likelihoods in a conditional model that generates each successive layer given its predecessor; i.e., $L_1(W\mathbf{x}, \phi) = -\log p_W(\phi|\mathbf{x})$ and $L_2(V\phi, \mathbf{y}) = -\log p_V(\mathbf{y}|\phi)$. (However, a loss based formulation is more flexible, since every negative log-likelihood is a loss but not vice versa.) Similarly to RBMs and probabilistic networks (PFNs) [27] (but unlike auto-encoders and classical feed-forward networks), we will not assume $\phi$ is a deterministic output of the first layer; instead we will consider $\phi$ to be a *variable* whose value is the subject of inference during training.

Given such a set-up many training principles become possible. For simplicity, we consider a Viterbi based training principle where the parameters $W$ and $V$ are optimized with respect to an optimal imputation of the latent values $\Phi$. To do so, define the first and second layer training objectives as

$$F_1(W, \Phi) = L_1(WX, \Phi) + \frac{\alpha}{2}\|W\|_F^2, \quad \text{and} \quad F_2(\Phi, V) = L_2(V\Phi, Y) + \frac{\beta}{2}\|V\|_F^2, \qquad (1)$$

where we assume the losses are convex in their first arguments. Here it is typical to assume that the losses decompose columnwise; that is, $L_1(\hat{\Psi}, \Phi) = \sum_{j=1}^{t} L_1(\hat{\psi}_j, \phi_j)$ and $L_2(Z, Y) = \sum_{j=1}^{t} L_2(\hat{\mathbf{z}}_j, \mathbf{y}_j)$, where $\hat{\psi}_j$ is the $j$th column of $\hat{\Psi}$ and $\hat{\mathbf{z}}_j$ is the $j$th column of $\hat{Z}$ respectively. This

follows for example if the training pairs $(\mathbf{x}_j, \mathbf{y}_j)$ are assumed I.I.D., but such a restriction is not necessary. Note that we have also introduced Euclidean regularization over the parameters (i.e. negative log-priors under a Gaussian), which will provide a useful representer theorem [28] we exploit later. These two objectives can be combined to obtain the following joint training problem:

$$\min_{W,V} \min_{\Phi} F_1(W, \Phi) + \gamma F_2(\Phi, V), \qquad (2)$$

where $\gamma > 0$ is a trade off parameter that balances the first versus second layer discrepancy. Unfortunately (2) is not jointly convex in the unknowns $W$, $V$ and $\Phi$.

A key modeling question concerns the structure of the latent representation $\phi$. As noted, the extensive literature on latent variable modeling has proposed a variety of forms for latent structure. Here, we follow work on deep learning and sparse coding and assume that the latent variables are boolean, $\phi \in \{0, 1\}^{h \times 1}$; an assumption that is also often made in auto-encoders [13], PFNs [27], and RBMs [5]. A boolean representation can capture structures that range from a single latent clustering [11, 19, 20], by imposing the assumption that $\phi'\mathbf{1} = 1$, to a general sparse code, by imposing the assumption that $\phi'\mathbf{1} = k$ for some small $k$ [1, 4, 13].[1] Observe that, in the latter case, one can control the complexity of the latent representation by imposing a constraint on the number of "active" variables $k$ rather than directly controlling the latent dimensionality $h$.

## 2.1 Multi-Layer Perceptrons and Large-Margin Losses

To complete a specification of the two-layer model in Figure 1 and the associated training problem (2), we need to commit to specific forms for the transfer functions $\mathbf{f}_1$ and $\mathbf{f}_2$ and the losses in (1). For simplicity, we will adopt a *large-margin* approach over two-layer *perceptrons*. Although it has been traditional in deep learning research to focus on exponential family conditional models (e.g. as in auto-encoders, PFNs and RBMs), these are not the only possibility; a large-margin approach offers additional sparsity and algorithmic simplifications that will clarify the development below. Despite its simplicity, such an approach will still be sufficient to prove our main point.

First, consider the second layer model. We will conduct our primary evaluations on multiclass classification problems, where output vectors $\mathbf{y}$ encode target classes by indicator vectors $\mathbf{y} \in \{0, 1\}^{m \times 1}$ such that $\mathbf{y}'\mathbf{1} = 1$. Although it is common to adopt a softmax transfer for $\mathbf{f}_2$ in such a case, it is also useful to consider a perceptron model defined by $\mathbf{f}_2(\hat{\mathbf{z}}) = \mathrm{indmax}(\hat{\mathbf{z}})$ such that $\mathrm{indmax}(\hat{\mathbf{z}}) = \mathbf{1}_i$ (vector of all 0s except a 1 in the $i$th position) where $\hat{z}_i \geq \hat{z}_l$ for all $l$. Therefore, for multi-class classification, we will simply adopt the standard large-margin multi-class loss [29]:

$$L_2(\hat{\mathbf{z}}, \mathbf{y}) = \max(\mathbf{1} - \mathbf{y} + \hat{\mathbf{z}} - \mathbf{1}\mathbf{y}'\hat{\mathbf{z}}). \qquad (3)$$

Intuitively, if $y_c = 1$ is the correct label, this loss encourages the response $\hat{z}_c = \mathbf{y}'\hat{\mathbf{z}}$ on the correct label to be a margin greater than the response $\hat{z}_i$ on any other label $i \neq c$.

Second, consider the first layer model. Although the loss (3) has proved to be highly successful for multi-class classification problems, it is not suitable for the first layer because it assumes there is only a *single* target component active in any latent vector $\phi$; i.e. $\phi'\mathbf{1} = 1$. Although some work has considered learning a latent *clustering* in a two-layer architecture [11, 18–20], such an approach is not able to capture the latent sparse code of a classical PFN or RBM in a reasonable way: using clustering to simulate a multi-dimensional sparse code causes exponential blow-up in the number of latent classes required. Therefore, we instead adopt a *multi-label* perceptron model for the first layer, defined by the transfer function $\mathbf{f}_1(\hat{\psi}) = \mathrm{step}(\hat{\psi})$ applied componentwise to the response vector $\hat{\psi}$; i.e. $\mathrm{step}(\hat{\psi}_i) = 1$ if $\hat{\psi}_i > 0$, 0 otherwise. Here again, instead of using a traditional negative log-likelihood loss, we will adopt a simple large-margin loss for multi-label classification that naturally accommodates multiple binary latent classifications in parallel. Although several loss formulations exist for multi-label classification [30, 31], we adopt the following:

$$L_1(\hat{\psi}, \phi) = \max(\mathbf{1} - \phi + \hat{\psi}\phi'\mathbf{1} - \mathbf{1}\phi'\hat{\psi}) \equiv \max\left((\mathbf{1} - \phi)/(\phi'\mathbf{1}) + \hat{\psi} - \mathbf{1}\phi'\hat{\psi}/(\phi'\mathbf{1})\right). (4)$$

Intuitively, this loss encourages the average response on the active labels, $\phi'\hat{\psi}/(\phi'\mathbf{1})$, to exceed the response $\hat{\psi}_i$ on any inactive label $i$, $\phi_i = 0$, by some margin, while also encouraging the response on any active label to match the average of the active responses. Despite their simplicity, large-margin multi-label losses have proved to be highly successful in practice [30, 31]. Therefore, the overall architecture we investigate embeds two nonlinear conditionals around a non-trivial latent layer.

## 3 Equivalent Reformulation

The main contribution of this paper is to show that the training problem (2) has a convex relaxation that preserves sufficient structure to transcend one-layer models. To demonstrate this relaxation, we first need to establish the key observation that problem (2) can be re-expressed in terms of a *kernel matrix* between latent representation vectors. Importantly, this reformulation allows the problem to be re-expressed in terms of an optimization objective that is jointly convex in all participating variables. We establish this key intermediate result in this section in three steps: first, by re-expressing the latent representation in terms of a latent kernel; second, by reformulating the second layer objective; and third, by reformulating the first layer objective by exploiting large-margin formulation outlined in Section 2.1. Below let $K = X'X$ denote the kernel matrix over the input data, let $\operatorname{Im}(N)$ denote the row space of $N$, and let and $\dagger$ denote Moore-Penrose pseudo-inverse.

First, simply define $N = \Phi'\Phi$. Next, re-express the second layer objective $F_2$ in (1) by the following.

**Lemma 1.** *For any fixed $\Phi$, letting $N = \Phi'\Phi$, it follows that*

$$\min_V F_2(\Phi, V) = \min_{B \in \operatorname{Im}(N)} L_2(B, Y) + \tfrac{\beta}{2} \operatorname{tr}(BN^\dagger B'). \tag{5}$$

*Proof.* The result follows from the following sequence of equivalence preserving transformations:

$$\min_V L_2(V\Phi, Y) + \tfrac{\beta}{2}\|V\|_F^2 = \min_A L_2(AN, Y) + \tfrac{\beta}{2} \operatorname{tr}(ANA') \tag{6}$$

$$= \min_{B \in \operatorname{Im}(N)} L_2(B, Y) + \tfrac{\beta}{2} \operatorname{tr}(BN^\dagger B'), \tag{7}$$

where, starting with the definition of $F_2$ in (1), the first equality in (6) follows from the representer theorem applied to $\|V\|_F^2$, which implies that the optimal $V$ must be in the form of $V = A\Phi'$ for some $A \in \mathbb{R}^{m \times t}$ [28]; and finally, (7) follows by the change of variable $B = AN$. $\square$

Note that Lemma 1 holds for any loss $L_2$. In fact, the result follows solely from the structure of the regularizer. However, we require $L_2$ to be convex in its first argument to ensure a convex problem below. Convexity is indeed satisfied by the choice (3). Moreover, the term $\operatorname{tr}(BN^\dagger B')$ is jointly convex in $N$ and $B$ since it is a perspective function [32], hence the objective in (5) is jointly convex.

Next, we reformulate the first layer objective $F_1$ in (1). Since this transformation exploits specific structure in the first layer loss, we present the result in two parts: first, by showing how the desired outcome follows from a general assumption on $L_1$, then demonstrating that this assumption is satisfied by the specific large-margin multi-label loss defined in (4). To establish this result we will exploit the following augmented forms for the data and variables: let $\tilde{\Phi} = [\Phi, kI]$, $\tilde{N} = \tilde{\Phi}'\tilde{\Phi}$, $\tilde{\Psi} = [\hat{\Psi}, 0]$, $\tilde{X} = [X, 0]$, $\tilde{K} = \tilde{X}'\tilde{X}$, and $\tilde{t} = t + h$.

**Lemma 2.** *For any $L_1$ if there exists a function $\tilde{L}_1$ such that $L_1(\hat{\Psi}, \Phi) = \tilde{L}_1(\tilde{\Phi}'\tilde{\Psi}, \tilde{\Phi}'\tilde{\Phi})$ for all $\hat{\Psi} \in \mathbb{R}^{h \times t}$ and $\Phi \in \{0, 1\}^{h \times t}$, such that $\Phi'\mathbf{1} = \mathbf{1}k$, it then follows that*

$$\min_W F_1(W, \Phi) = \min_{D \in \operatorname{Im}(\tilde{N})} \tilde{L}_1(D\tilde{K}, \tilde{N}) + \tfrac{\alpha}{2} \operatorname{tr}(D'\tilde{N}^\dagger D\tilde{K}). \tag{8}$$

*Proof.* Similar to above, consider the sequence of equivalence preserving transformations:

$$\min_W L_1(WX, \Phi) + \tfrac{\alpha}{2}\|W\|_F^2 = \min_W \tilde{L}_1(\tilde{\Phi}'W\tilde{X}, \tilde{\Phi}'\tilde{\Phi}) + \tfrac{\alpha}{2}\|W\|_F^2 \tag{9}$$

$$= \min_C \tilde{L}_1(\tilde{\Phi}'\tilde{\Phi}C\tilde{X}'\tilde{X}, \tilde{\Phi}'\tilde{\Phi}) + \tfrac{\beta}{2} \operatorname{tr}(\tilde{X}C'\tilde{\Phi}'\tilde{\Phi}C\tilde{X}') \tag{10}$$

$$= \min_{D \in \operatorname{Im}(\tilde{N})} \tilde{L}_1(D\tilde{K}, \tilde{N}) + \tfrac{\alpha}{2} \operatorname{tr}(D'\tilde{N}^\dagger D\tilde{K}), \tag{11}$$

where, starting with the definition of $F_1$ in (1), the first equality (9) simply follows from the assumption. The second equality (10) follows from the representer theorem applied to $\|W\|_F^2$, which implies that the optimal $W$ must be in the form of $W = \tilde{\Phi}C\tilde{X}'$ for some $C \in \mathbb{R}^{\tilde{t} \times \tilde{t}}$ (using the fact that $\tilde{\Phi}$ has full rank $h$) [28]. Finally, (11) follows by the change of variable $D = \tilde{N}C$. $\square$

Observe that the term $\text{tr}(D'\tilde{N}^\dagger D\tilde{K})$ is again jointly convex in $\tilde{N}$ and $D$ (also a perspective function), while it is easy to verify that $\tilde{L}_1(D\tilde{K},\tilde{N})$ as defined in Lemma 3 below is also jointly convex in $\tilde{N}$ and $D$ [32]; therefore the objective in (8) is jointly convex.

Next, we show that the assumption of Lemma 2 is satisfied by the specific large-margin multi-label formulation in Section 2.1; that is, assume $L_1$ is given by the large-margin multi-label loss (4):

$$
\begin{aligned}
L_1(\hat{\Psi},\Phi) &= \sum_j \max\left(\mathbf{1} - \phi_j + \hat{\psi}_j\phi_j'\mathbf{1} - \mathbf{1}\phi_j'\hat{\psi}_j\right)\\
&= \tau\left(\mathbf{11}' - \Phi + \hat{\Psi}\,\text{diag}(\Phi'\mathbf{1}) - \mathbf{1}\,\text{diag}(\Phi'\hat{\Psi})'\right), \quad \text{such that } \tau(\Theta) := \sum_j \max(\boldsymbol{\theta}_j), \;(12)
\end{aligned}
$$

where we use $\hat{\psi}_j$, $\phi_j$ and $\boldsymbol{\theta}_j$ to denote the $j$th columns of $\hat{\Psi}$, $\Phi$ and $\Theta$ respectively.

**Lemma 3.** *For the multi-label loss $L_1$ defined in (4), and for any fixed $\Phi \in \{0,1\}^{h\times t}$ where $\Phi'\mathbf{1} = \mathbf{1}k$, the definition $\tilde{L}_1(\tilde{\Phi}'\tilde{\Psi}, \tilde{\Phi}'\tilde{\Phi}) := \tau(\tilde{\Phi}'\tilde{\Psi} - \tilde{\Phi}'\tilde{\Phi}/k) + t - \text{tr}(\tilde{\Phi}'\tilde{\Psi})$ using the augmentation above satisfies the property that $L_1(\hat{\Psi},\Phi) = \tilde{L}_1(\tilde{\Phi}'\tilde{\Psi}, \tilde{\Phi}'\tilde{\Phi})$ for any $\hat{\Psi} \in \mathbb{R}^{h\times t}$.*

*Proof.* Since $\Phi'\mathbf{1} = \mathbf{1}k$ we obtain a simplification of $L_1$:

$$
L_1(\hat{\Psi},\Phi) = \tau\left(\mathbf{11}' - \Phi + k\hat{\Psi} - \mathbf{1}\,\text{diag}(\Phi'\hat{\Psi})'\right) = \tau(k\hat{\Psi} - \Phi) + t - \text{tr}(\tilde{\Phi}'\tilde{\Psi}). \quad (13)
$$

It only remains is to establish that $\tau(k\hat{\Psi} - \Phi) = \tau(\tilde{\Phi}'\tilde{\Psi} - \tilde{\Phi}'\tilde{\Phi}/k)$. To do so, consider the sequence of equivalence preserving transformations:

$$
\tau(k\hat{\Psi} - \Phi) = \max_{\Lambda \in \mathbb{R}_+^{h\times\tilde{t}}:\Lambda'\mathbf{1}=\mathbf{1}} \text{tr}\left(\Lambda'(k\tilde{\Psi} - \tilde{\Phi})\right) \quad (14)
$$

$$
= \max_{\Omega \in \mathbb{R}_+^{\tilde{t}\times\tilde{t}}:\Omega'\mathbf{1}=\mathbf{1}} \tfrac{1}{k}\,\text{tr}\left(\Omega'\tilde{\Phi}'(k\tilde{\Psi} - \tilde{\Phi})\right) = \tau(\tilde{\Phi}'\tilde{\Psi} - \tilde{\Phi}'\tilde{\Phi}/k), \quad (15)
$$

where the equalities in (14) and (15) follow from the definition of $\tau$ and the fact that linear maximizations over the simplex obtain their solutions at the vertices. To establish the equality between (14) and (15), since $\tilde{\Phi}$ embeds the submatrix $kI$, for any $\Lambda \in \mathbb{R}_+^{h\times\tilde{t}}$ there must exist an $\Omega \in \mathbb{R}_+^{\tilde{t}\times\tilde{t}}$ satisfying $\Lambda = \tilde{\Phi}\Omega/k$. Furthermore, these matrices satisfy $\Lambda'\mathbf{1} = \mathbf{1}$ iff $\Omega'\tilde{\Phi}'\mathbf{1}/k = \mathbf{1}$ iff $\Omega'\mathbf{1} = \mathbf{1}$. $\quad\square$

Therefore, the result (8) holds for the first layer loss (4), using $\tilde{L}_1$ defined in Lemma 3. (The same result can be established for other loss functions, such as the multi-class large-margin loss.) Combining these lemmas yields the desired result of this section.

**Theorem 1.** *For any second layer loss and any first layer loss that satisfies the assumption of Lemma 2 (for example the large-margin multi-label loss (4)), the following equivalence holds:*

$$
(2) = \min_{\{\tilde{N}:\exists\Phi\in\{0,1\}^{t\times h}\,s.t.\;\Phi\mathbf{1}=\mathbf{1}k,\tilde{N}=\tilde{\Phi}'\tilde{\Phi}\}} \min_{B\in\text{Im}(\tilde{N})} \min_{D\in\text{Im}(\tilde{N})} \tilde{L}_1(D\tilde{K},\tilde{N}) + \tfrac{\alpha}{2}\text{tr}(D'\tilde{N}^\dagger D\tilde{K})
$$

$$
+ \gamma L_2(B,Y) + \tfrac{\gamma\beta}{2}\text{tr}(B\tilde{N}^\dagger B'). \quad (16)
$$

(Theorem 1 follows immediately from Lemmas 1 and 2.) Note that no relaxation has occurred thus far: the objective value of (16) matches that of (2). Not only has this reformulation resulted in (2) being entirely expressed in terms of the latent kernel matrix $\tilde{N}$, the objective in (16) is jointly convex in all participating unknowns, $\tilde{N}$, $B$ and $D$. Unfortunately, the *constraints* in (16) are not convex.

## 4 Convex Relaxation

We first relax the problem by dropping the augmentation $\Phi \mapsto \tilde{\Phi}$ and working with the $t \times t$ variable $N = \Phi'\Phi$. Without the augmentation, Lemma 3 becomes a lower bound (i.e. (14)$\geq$(15)), hence a relaxation. To then achieve a convex form we further relax the constraints in (16). To do so, consider

$$
\mathcal{N}_0 = \left\{N : \exists\Phi \in \{0,1\}^{t\times h} \text{ such that } \Phi\mathbf{1} = \mathbf{1}k \text{ and } N = \Phi'\Phi\right\} \quad (17)
$$

$$
\mathcal{N}_1 = \left\{N : N \in \{0,...,k\}^{t\times t}, N \succeq 0, \text{diag}(N) = \mathbf{1}k, \text{rank}(N) \leq h\right\} \quad (18)
$$

$$
\mathcal{N}_2 = \left\{N : N \geq 0, N \succeq 0, \text{diag}(N) = \mathbf{1}k\right\}, \quad (19)
$$

where it is clear from the definitions that $\mathcal{N}_0 \subseteq \mathcal{N}_1 \subseteq \mathcal{N}_2$. (Here we use $N \succeq 0$ to also encode $N' = N$.) Note that the set $\mathcal{N}_0$ corresponds to the original set of constraints from (16). The set

---

**Algorithm 1:** ADMM to optimize $\mathcal{F}(N)$ for $N \in \mathcal{N}_2$.

---
1  Initialize: $M_0 = I, \Gamma_0 = \mathbf{0}$.
2  **while** $T = 1, 2, \ldots$ **do**
3     |  $N_T \leftarrow \arg\min_{N \succeq 0} \mathcal{L}(N, M_{T-1}, \Gamma_{T-1})$, by using the boosting Algorithm 2.
4     |  $M_T \leftarrow \arg\min_{M \geq 0, M_{ii}=k} \mathcal{L}(N_T, M, \Gamma_{T-1})$, which has an efficient closed form solution.
5     |  $\Gamma_T \leftarrow \Gamma_{T-1} + \frac{1}{\mu}(M_T - N_T)$; i.e. update the multipliers.
6  **return** $N_T$.

---

---

**Algorithm 2:** Boosting algorithm to optimize $\mathcal{G}(N)$ for $N \succeq 0$.

---
1  Initialize: $N_0 \leftarrow \mathbf{0}$, $H_0 \leftarrow [\,]$ (empty set).
2  **while** $T = 1, 2, \ldots$ **do**
3     |  Find the smallest arithmetic eigenvalue of $\nabla\mathcal{G}(N_{T-1})$, and its eigenvector $\mathbf{h}_T$.
4     |  Conic search by LBFGS: $(a_T, b_T) \leftarrow \min_{a \geq 0, b \geq 0} \mathcal{G}(aN_{T-1} + b\mathbf{h}_T\mathbf{h}_T')$.
      |  Local search by LBFGS: $H_T \leftarrow \text{local\_min}_H \mathcal{G}(HH')$ initialized by $H = (\sqrt{a}H_{T-1}, \sqrt{b}\mathbf{h}_T)$.
5     |  Set $N_T \leftarrow H_T H_T'$; **break** if stopping criterion met.
6  **return** $N_T$.

---

$\mathcal{N}_1$ simplifies the characterization of this constraint set on the resulting kernel matrices $N = \Phi'\Phi$. However, neither $\mathcal{N}_0$ nor $\mathcal{N}_1$ are convex. Therefore, we need to adopt the further relaxed set $\mathcal{N}_2$, which is convex. (Note that $N_{ij} \leq k$ has been implied by $N \succeq 0$ and $N_{ii} = k$ in $\mathcal{N}_2$.) Since dropping the rank constraint eliminates the constraints $B \in \text{Im}(N)$ and $D \in \text{Im}(N)$ in (16) when $N \succ 0$ [32], we obtain the following relaxed problem, which is jointly convex in $N$, $B$ and $D$:

$$\min_{N \in \mathcal{N}_2} \min_{B \in \mathbb{R}^{t \times t}} \min_{D \in \mathbb{R}^{t \times t}} \tilde{L}_1(DK, N) + \frac{\alpha}{2}\text{tr}(D'N^\dagger DK) + \gamma L_2(B, Y) + \frac{\gamma\beta}{2}\text{tr}(BN^\dagger B'). \qquad (20)$$

## 5   Efficient Training Approach

Unfortunately, nonlinear semidefinite optimization problems in the form (20) are generally thought to be too expensive in practice despite their polynomial theoretical complexity [33, 34]. Therefore, we develop an effective training algorithm that exploits problem structure to bypass the main computational bottlenecks. The key challenge is that $\mathcal{N}_2$ contains both semidefinite and affine constraints, and the pseudo-inverse $N^\dagger$ makes optimization over $N$ difficult even for fixed $B$ and $D$.

To mitigate these difficulties we first treat (20) as the reduced problem, $\min_{N \in \mathcal{N}_2} \mathcal{F}(N)$, where $\mathcal{F}$ is an implicit objective achieved by minimizing out $B$ and $D$. Note that $\mathcal{F}$ is still convex in $N$ by the joint convexity of (20). To cope with the constraints on $N$ we adopt the alternating direction method of multipliers (ADMM) [35] as the main outer optimization procedure; see Algorithm 1. This approach allows one to divide $\mathcal{N}_2$ into two groups, $N \succeq 0$ and $\{N_{ij} \geq 0, N_{ii} = k\}$, yielding the augmented Lagrangian

$$\mathcal{L}(N, M, \Gamma) = \mathcal{F}(N) + \delta(N \succeq 0) + \delta(M_{ij} \geq 0, M_{ii} = k) - \langle \Gamma, N - M \rangle + \frac{1}{2\mu}\|N - M\|_F^2, \quad (21)$$

where $\mu > 0$ is a small constant, and $\delta$ denotes an indicator such that $\delta(\cdot) = 0$ if $\cdot$ is true, and $\infty$ otherwise. In this procedure, Steps 4 and 5 cost $O(t^2)$ time; whereas the main bottleneck is Step 3, which involves minimizing $\mathcal{G}_T(N) := \mathcal{L}(N, M_{T-1}, \Gamma_{T-1})$ over $N \succeq 0$ for fixed $M_{T-1}$ and $\Gamma_{T-1}$.

**Boosting for Optimizing over the Positive Semidefinite Cone.** To solve the problem in Step 3 we develop an efficient boosting procedure based on [36] that retains low rank iterates $N_T$ while avoiding the need to determine $N^\dagger$ when computing $\mathcal{G}(N)$ and $\nabla\mathcal{G}(N)$; see Algorithm 2. The key idea is to use a simple change of variable. For example, consider the first layer objective and let $\mathcal{G}_1(N) = \min_D \tilde{L}_1(DK, N) + \frac{\alpha}{2}\text{tr}(D'N^\dagger DK)$. By defining $D = NC$, we obtain $\mathcal{G}_1(N) = \min_C \tilde{L}_1(NCK, N) + \frac{\alpha}{2}\text{tr}(C'NCK)$, which no longer involves $N^\dagger$ but remains convex in $C$; this problem can be solved efficiently after a slight smoothing of the objective [37] (e.g. by LBFGS). Moreover, the gradient $\nabla\mathcal{G}_1(N)$ can be readily computed given $C^*$. Applying the same technique

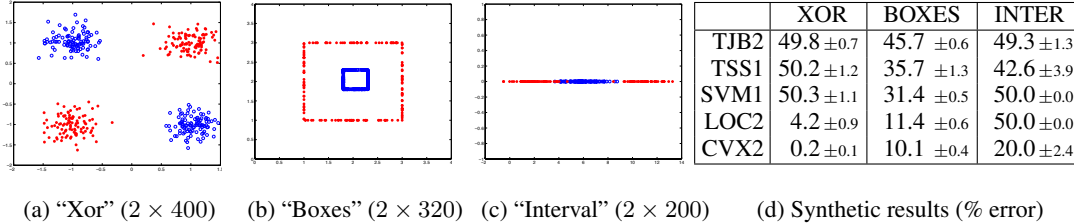

|  | XOR | BOXES | INTER |
|---|---|---|---|
| TJB2 | 49.8 $_{\pm0.7}$ | 45.7 $_{\pm0.6}$ | 49.3 $_{\pm1.3}$ |
| TSS1 | 50.2 $_{\pm1.2}$ | 35.7 $_{\pm1.3}$ | 42.6 $_{\pm3.9}$ |
| SVM1 | 50.3 $_{\pm1.1}$ | 31.4 $_{\pm0.5}$ | 50.0 $_{\pm0.0}$ |
| LOC2 | 4.2 $_{\pm0.9}$ | 11.4 $_{\pm0.6}$ | 50.0 $_{\pm0.0}$ |
| CVX2 | 0.2 $_{\pm0.1}$ | 10.1 $_{\pm0.4}$ | 20.0 $_{\pm2.4}$ |

(a) "Xor" ($2 \times 400$)  (b) "Boxes" ($2 \times 320$)  (c) "Interval" ($2 \times 200$)  (d) Synthetic results (% error)

Figure 2: Synthetic experiments: three artificial data sets that cannot be meaningfully classified by a one-layer model that does not use a nonlinear kernel. Table shows percentage test set error.

to the second layer yields an efficient procedure for evaluating $\mathcal{G}(N)$ and $\nabla\mathcal{G}(N)$. Finally note that many of the matrix-vector multiplications in this procedure can be further accelerated by exploiting the low rank factorization of $N$ maintained by the boosting algorithm; see the Appendix for details.

**Additional Relaxation.** One can further reduce computation cost by adopting additional relaxations to (20). For example, by dropping $N \geq 0$ and relaxing $\mathrm{diag}(N) = \mathbf{1}k$ to $\mathrm{diag}(N) \leq \mathbf{1}k$, the objective can be written as $\min_{\{N \succeq 0, \max_i N_{ii} \leq k\}} \mathcal{F}(N)$. Since $\max_i N_{ii}$ is convex in $N$, it is well known that there must exist a constant $c_1 > 0$ such that the optimal $N$ is also an optimal solution to $\min_{N \succeq 0} \mathcal{F}(N) + c_1 \left(\max_i N_{ii}\right)^2$. While $\max_i N_{ii}$ is not smooth, one can further smooth it with a softmax, to instead solve $\min_{N \succeq 0} \mathcal{F}(N) + c_1 \left(\log \sum_i \exp(c_2 N_{ii})\right)^2$ for some large $c_2$. This formulation avoids the need for ADMM entirely and can be directly solved by Algorithm 2.

# 6 Experimental Evaluation

To investigate the effectiveness of the proposed relaxation scheme for training a two-layer conditional model, we conducted a number of experiments to compare learning quality against baseline methods. Note that, given an optimal solution $N$, $B$ and $D$ to (20), an approximate solution to the original problem (2) can be recovered heuristically by first rounding $N$ to obtain $\Phi$, then recovering $W$ and $V$, as shown in Lemmas 1 and 2. However, since our primary objective is to determine whether *any* convex relaxation of a two-layer model can even compete with one-layer or locally trained two-layer models (rather than evaluate heuristic rounding schemes), we consider a transductive evaluation that does not require any further modification of $N$, $B$ and $D$. In such a set-up, training data is divided into a labeled and unlabeled portion, where the method receives $X = [X_\ell, X_u]$ and $Y_\ell$, and at test time the resulting predictions $\hat{Y}_u$ are evaluated against the held-out labels $Y_u$.

**Methods.** We compared the proposed convex relaxation scheme (CVX2) against the following methods: simple alternating minimization of the same two-layer model (2) (LOC2), a one-layer linear SVM trained on the labeled data (SVM1), the transductive one-layer SVM methods of [38] (TSJ1) and [39] (TSS1), and the transductive latent clustering method of [18, 19] (TJB2), which is also a two-layer model. Linear input kernels were used for all methods (standard in most deep learning models) to control the comparison between one and two-layer models. Our experiments were conducted with the following common protocol: First, the data was split into a separate training and test set. Then the parameters of each procedure were optimized by a three-fold cross validation on the training set. Once the optimal parameters were selected, they were fixed and used on the test set. For transductive procedures, the same three training sets from the first phase were used, but then combined with ten new test sets drawn from the disjoint test data (hence 30 overall) for the final evaluation. At no point were test examples used to select any parameters for any of the methods. We considered different proportions between labeled/unlabeled data; namely, 100/100 and 200/200.

**Synthetic Experiments.** We initially ran a proof of concept experiment on three binary labeled artificial data sets depicted in Figure 2 (showing data set sizes $n \times t$) with 100/100 labeled/unlabeled training points. Here the goal was simply to determine whether the relaxed two-layer training method could preserve sufficient structure to overcome the limits of a one-layer architecture. Clearly, none of the data sets in Figure 2 are adequately modeled by a one-layer architecture (that does not cheat and use a nonlinear kernel). The results are shown in the Figure 2(d) table.

|       | MNIST | USPS | Letter | COIL | CIFAR | G241N |
|-------|-------|------|--------|------|-------|-------|
| TJB2  | $19.3_{\pm1.2}$ | $53.2_{\pm2.9}$ | $20.4_{\pm2.1}$ | $30.6_{\pm0.8}$ | $29.2_{\pm2.1}$ | $26.3_{\pm0.8}$ |
| LOC2  | $19.3_{\pm1.0}$ | $13.9_{\pm1.1}$ | $10.4_{\pm0.6}$ | $18.0_{\pm0.5}$ | $31.8_{\pm0.9}$ | $41.6_{\pm0.9}$ |
| SVM1  | $16.2_{\pm0.7}$ | $11.6_{\pm0.5}$ | $6.2_{\pm0.4}$ | $16.9_{\pm0.6}$ | $27.6_{\pm0.9}$ | $27.1_{\pm0.9}$ |
| TSS1  | $13.7_{\pm0.8}$ | $11.1_{\pm0.5}$ | $5.9_{\pm0.5}$ | $17.5_{\pm0.6}$ | $26.7_{\pm0.7}$ | $25.1_{\pm0.8}$ |
| TSJ1  | $14.6_{\pm0.7}$ | $12.1_{\pm0.4}$ | $5.6_{\pm0.5}$ | $17.2_{\pm0.6}$ | $26.6_{\pm0.8}$ | $24.4_{\pm0.7}$ |
| CVX2  | $9.2_{\pm0.6}$ | $9.2_{\pm0.5}$ | $5.1_{\pm0.5}$ | $13.8_{\pm0.6}$ | $26.5_{\pm0.8}$ | $25.2_{\pm1.0}$ |

Table 1: Mean test misclassification error % ($\pm$ stdev) for 100/100 labeled/unlabeled.

|       | MNIST | USPS | Letter | COIL | CIFAR | G241N |
|-------|-------|------|--------|------|-------|-------|
| TJB2  | $13.7_{\pm0.6}$ | $46.6_{\pm1.0}$ | $14.0_{\pm2.6}$ | $45.0_{\pm0.8}$ | $30.4_{\pm1.9}$ | $22.4_{\pm0.5}$ |
| LOC2  | $16.3_{\pm0.6}$ | $9.7_{\pm0.5}$ | $8.5_{\pm0.6}$ | $12.8_{\pm0.6}$ | $28.2_{\pm0.9}$ | $40.4_{\pm0.7}$ |
| SVM1  | $11.2_{\pm0.4}$ | $10.7_{\pm0.4}$ | $5.0_{\pm0.3}$ | $15.6_{\pm0.5}$ | $25.5_{\pm0.6}$ | $22.9_{\pm0.5}$ |
| TSS1  | $11.4_{\pm0.5}$ | $11.3_{\pm0.4}$ | $4.4_{\pm0.3}$ | $14.9_{\pm0.4}$ | $24.0_{\pm0.6}$ | $23.7_{\pm0.5}$ |
| TSJ1  | $12.3_{\pm0.5}$ | $11.8_{\pm0.4}$ | $4.8_{\pm0.3}$ | $13.5_{\pm0.4}$ | $23.9_{\pm0.5}$ | $22.2_{\pm0.6}$ |
| CVX2  | $8.8_{\pm0.4}$ | $6.6_{\pm0.4}$ | $3.8_{\pm0.3}$ | $8.2_{\pm0.4}$ | $22.8_{\pm0.6}$ | $20.3_{\pm0.5}$ |

Table 2: Mean test misclassification error % ($\pm$ stdev) for 200/200 labeled/unlabeled.

As expected, the one-layer models SVM1 and TSS1 were unable to capture any useful classification structure in these problems. (TSJ1 behaves similarly to TSS1.) The results obtained by CVX2, on the other hand, are encouraging. In these data sets, CVX2 is easily able to capture latent nonlinearities while outperforming the locally trained LOC2. Although LOC2 is effective in the first two cases, it exhibits weaker test accuracy while failing on the third data set. The two-layer method TJB2 exhibited convergence difficulties on these problems that prevented reasonable results.

**Experiments on "Real" Data Sets.** Next, we conducted experiments on real data sets to determine whether the advantages in controlled synthetic settings could translate into useful results in a more realistic scenario. For these experiments we used a collection of binary labeled data sets: USPS, COIL and G241N from [40], Letter from [41], MNIST, and CIFAR-100 from [42]. (See Appendix B in the supplement for further details.) The results are shown in Tables 1 and 2 for the labeled/unlabeled proportions 100/100 and 200/200 respectively.

The relaxed two-layer method CVX2 again demonstrates effective results, although some data sets caused difficulty for all methods. The data sets can be divided into two groups, (MNIST, USPS, COIL) versus (Letter, CIFAR, G241N). In the first group, two-layer modeling demonstrates a clear advantage: CVX2 outperforms SVM1 by a significant margin. Note that this advantage must be due to two-layer versus one-layer modeling, since the transductive SVM methods TSS1 and TSJ1 demonstrate no advantage over SVM1. For the second group, the effectiveness of SVM1 demonstrates that only minor gains can be possible via transductive or two-layer extensions, although some gains are realized. The locally trained two-layer model LOC2 performed quite poorly in all cases. Unfortunately, the convex latent clustering method TJB2 was also not competitive on any of these data sets. Overall, CVX2 appears to demonstrate useful promise as a two-layer modeling approach.

## 7  Conclusion

We have introduced a new convex approach to two-layer conditional modeling by reformulating the problem in terms of a *latent kernel* over intermediate feature representations. The proposed model can accommodate latent feature representations that go well beyond a latent clustering, extending current convex approaches. A semidefinite relaxation of the latent kernel allows a reasonable implementation that is able to demonstrate advantages over single-layer models and local training methods. From a deep learning perspective, this work demonstrates that trainable latent layers can be expressed in terms of reproducing kernel Hilbert spaces, while large margin methods can be usefully applied to multi-layer prediction architectures. Important directions for future work include replacing the step and indmax transfers with more traditional sigmoid and softmax transfers, while also replacing the margin losses with more traditional Bregman divergences; refining the relaxation to allow more control over the structure of the latent representations; and investigating the utility of convex methods for stage-wise training within multi-layer architectures.

## Footnotes

[1] Throughout this paper we let $\mathbf{1}$ denote the vector of all 1s with length determined by context.

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
