[Supplementary Material · Convex Two-Layer Modeling-supplement.pdf]

# A  Additional Algorithm Details

Let us analyze and optimize the computational cost for updates in the boosting algorithm 2. $N_T \leftarrow H_T H'_T$ in Step 5 is just conceptual. The bottleneck is the conic/local search in Step 4, and the computation of $\nabla\mathcal{G}(N_{T-1})$ in Step 3. Here we will show that the whole boosting algorithm can be run by only using $H_T$, and thanks for this explicit decomposition of $N_T$, the gradient of $\mathcal{G}$ in both $N$ and $H$ can be computed considerably faster than just having $N_T$.

## A.1  Faster Gradient Calculation by Low Rank Decomposition

Recall that $\mathcal{G}$ consists mainly of two parts: the first layer objective in (11) and the second layer objective in (7). We just show how to solve the first layer, while the technique can be applied directly to the second layer too. For simplicity, we only consider linear kernel on $X$, and extension to nonlinear kernel is also straightforward.

For convenience we copy (11) to here, using $N_T$ and $H_T$:

$$\min_{W} L_1(WX, H_T) + \tfrac{\alpha}{2}\|W\|_F^2 = \min_{W} \tilde{L}_1(H'_T WX, H'_T H_T) + \tfrac{\alpha}{2}\|W\|_F^2 \tag{22}$$

$$= \min_{C} \tilde{L}_1(H'_T H_T CX'X, H'_T H_T) + \tfrac{\beta}{2}\operatorname{tr}(XC'H'_T H_T CX') \tag{23}$$

$$= \min_{D \in \operatorname{Im}(N_T)} \tilde{L}_1(DK, N_T) + \tfrac{\alpha}{2}\operatorname{tr}(D'N_T^\dagger DK). \tag{24}$$

Denote this objective as $\mathcal{G}_1(N)$ or $\mathcal{H}_1(H)$. The boosting Step 4 indeed only requires the gradient $\nabla\mathcal{H}_1(H_T)$, while the gradient $\nabla\mathcal{G}_1(N_T)$ is needed only in Step 3. So we focus on the efficient computation of these two gradients.

To compute $\nabla\mathcal{H}_1(H_T)$, it suffices to optimize over $W$ in (22). This is advantageous because a) the objective is strongly convex which is in favor of LBFGS; b) the size of $W$ is $nT$, where $T$ is the iteration index of boosting and is often quite small; c) the gradient in $W$ can be computed in $O(tnT)$ time, which can also benefit from the low value of $T$.

To compute $\nabla\mathcal{G}_1(N_T)$, one possible approach is to solve for $C$ in (23):

$$\min_{C} \tilde{L}_1(N_T CX'X, N_T) + \tfrac{\beta}{2}\operatorname{tr}(XC'N_T CX'). \tag{25}$$

However, the cost for computing the gradient in $C$ is $O(t^2 n)$, which is expensive if done at each iteration of optimization. Therefore we introduce one more change of variable: $E = CX'$ and then the problem becomes

$$\min_{E} \tilde{L}_1(N_T EX, N_T) + \tfrac{\beta}{2}\operatorname{tr}(E'N_T E). \tag{26}$$

So our final strategy is:

- Find the optimal $W^*$ for (22) as in computing $\nabla\mathcal{H}_1(H_T)$,
- Recover the optimal $E^*$ for (26) by finding any $E$ that satisfies $W^* = H_T E$,
- Use $E^*$ to compute the gradient $\nabla\mathcal{G}_1(N_T)$ via (26).

The first and second steps can make use of the low value of $T$ as in computing $\nabla\mathcal{H}_1(H_T)$. The last step does cost $O(t^2 n)$, but it needs to be done only once, rather than in each iteration of solving for $C$ in (25). So in summary, the total computational cost is $O(tnT)$ per iteration in optimizing $W$, followed by $O(t^2 n)$ for one time to recover $E^*$ and to compute $\nabla\mathcal{G}_1(N_T)$ via (26).

# B  Additional details on the experimental data

For the "real" experiments we used a collection of binary labeled data sets: USPS ($241 \times 1500$) and G241N ($241 \times 1500$) from [40], Letter (vowel letters A-E vs non vowel letters B-F $16 \times 3098$) from [41], MNIST ($\{1, 9\}$vs$\{4, 8\}$: $784 \times 28484$), and CIFAR-100 (bicycle and motorcycle vs lawnmower and tank $256 \times 1526$ where red channel features are preprocessed by averaging pixels) from [42].