[Reviews · NeurIPS 2013]

Submitted by Assigned_Reviewer_5

The paper addresses problem of training two-layer network for classification with latent variables. The authors propose a convex SDP relaxation of originally non-convex training problem. They also provide an approximate optimization algorithm to solve their SDP formulation. A proof of concept experiments show promising results, namely, that the algorithm outperforms both globally optimized single-layer models as well as the same two-layer model optimized with local alternating minimization.

The proposed reformulation which allows SDP relaxation is interesting and
novel. Overall the paper is sufficiently clear though some parts of the text are
dense. The paper seems to be technically sound.

The main weakness seems to be complexity of the resulting SDP problem (21). The
authors could mention basic properties of the proposed optimization algorithm
for solving (21), e.g computational time required for the benchmark problems and
whether the algorithm provides a precise solution (i.e. what was the stopping
condition used). This is an important information because convex problem does
not immediately mean easy to solve problem, i.e. a convex relaxation can be
intractable in practice and it should be clear if it is the case or not. However, the proposed relaxation would be valuable even in this case.

Minor comments:
- equ (17): variable is missing below the first minimum
- equ (18): I think N=\Phi'*\Phi should appear in the conditions defining the set.
- line 331: (d) Synhetic results
- line 356: It is unclear why the used transductive evaluation does not require
computing responses of the network and thus knowledge of W and V.
- I could not find the actual size of instances of the problem (21) solved
in the experiments.
Summary: The proposed SDP relaxation is an interesting attempt to find a better
approximation of an important instance of non-convex training problems. Though
the current algorithm may not be practical it can inspire development of more
efficient methods.

Submitted by Assigned_Reviewer_6

This paper presents a convex approach to train a two-layer model in supervised learning. This is achieved by incorporating large margin losses in the training objectives, adopting indmax transfer for the second layer and multi-label perception model with a step transfer for the first layer; finally, convex relaxations are applied that make global training of a two-layer model possible.

The paper is well written, and the process of linking all the factors together and deriving a convex objective is interesting and insightful. The authors did a good job presenting the technical steps taken to the ultimate convex formulation, with some nice intermediate results. The global training methodology proposed in this paper is meaningful from a deep learning perspective, and experimental results are convincing to demonstrate the significance of global training.
Summary: The paper presents a convex approach to train a conditional two-layer model. The technical work in this paper is solid, while the conclusion is significant and insightful.

Submitted by Assigned_Reviewer_7

A convex relaxation of two-layer neural network is proposed in the
paper. This paper is well-written. The experiments show good
performance on the "real" datasets. But my main concern is the
scalability of this approach.

The approach of using SDP for convex relaxation was widely used more
than 10 years ago, nothing new here. Though it has a nice form, the
scalability is the major issue of this type of relaxation. Here, we
need to optimize a problem of t^2, the square of the size of
instances, which is probably only feasible for toy datasets. For the scalability issue, it would be better to compare the training time among algorithms.

Algorithm 2 takes advantage of low rank of N. However, the rank of N
is not guaranteed to be small.

For those synthetic experiments, RBF SVM probably can achieve a quite
good. A fair comparison would be Nystrom approximation to RBF SVM with
random selected bases, instead of one-layer linear SVM.
Summary: Well-written. But SDP convex relaxation is not novel.
Author Feedback

Author rebuttal: Thanks to the reviewers for their comments.



Assigned_Reviewer_5:

As requested, the average training times (in seconds) of the various two-layer methods are as follows:

Experiment 1: train/test = 100/100, average training time:
MNIST: CVX2 = 38s (21), LOC2 = 4s, TJB2 = 18s
USPS: CVX2 = 43s (23), LOC2 = 14s, TJB2 = 8s
Letter: CVX2 = 32s (26), LOC2 = 191s, TJB2 = 3054s (see Note)
CIFAR: CVX2 = 10s (21), LOC2 = 6s, TJB2 = 15s
G241N: CVX2 = 14s (24), LOC2 = 11s, TJB2 = 4s
(Average rank of the CVX2 solutions are shown in brackets.)

Experiment 2: train/test = 200/200, average training time:
MNIST: CVX2 = 158s (22), LOC2 = 85s, TJB2 = 162s
USPS: CVX2 = 224s (25), LOC2 = 201s, TJB2 = 3790s (see Note)
Letter: CVX2 = 101s (24), LOC2 = 180s, TJB2 = 66s
CIFAR: CVX2 = 94s (24), LOC2 = 51s, TJB2 = 118s
G241N: CVX2 = 260s (34), LOC2 = 21s, TJB2 = 24s
(Average rank of the CVX2 solutions are shown in brackets.)

Recall that in the implementation of the proposed convex method, CVX2, each boosting step (which adds a single rank to the solution) is interleaved with local optimization. For the local optimization we use a standard LBFGS implementation with default termination conditions. For the outer boosting iterations we terminate when the relative objective improvement is less than 5e-5 or the absolute improvement is less than 1e-3. The average rank results in the above table corresponds to the number of boosting rounds used by CVX2, which also determines the rank of its final solutions. From these results, one can see that the method uses significantly less than the full O(t^2) storage.

Note: We obtained the implementation of TJB2 from [18]. Unfortunately, that implementation experiences convergence issues on certain problems and settings, and we were not able to rectify the problem in the obtained code.

Addressing the minor comments:

- Thanks
- Yes
- Thanks
- Yes, the transductive evaluation needs to use V and Phi. More precisely, it uses A and N, since V Phi = A N as given in Lemma 1.
- The experiments involve instances of (21) of two sizes: t = 200 (in the 100/100 train/test split cases) and t = 400 (in the 200/200 train/test split cases).



Assigned_Reviewer_6:

Thanks



Assigned_Reviewer_7:

As the paper explains in Section 5, the boosting algorithm developed for CVX2 is explicitly designed to avoid optimizing over variables of t^2 size. The experimental results (given above in the response to Assigned_Reviewer_5) demonstrate that the ranks of the final solutions returned by CVX2 remain very small compared to the size of the training data. In practice, the boosting implementation of CVX2 tends to use O(t) rather than Omega(t^2) space because it terminates with small rank. The experimental results also show that the training times of the method are not significantly worse than the other two-layer methods on these problems.

Although semidefinite relaxation has been applied to other machine learning problems in the past, no such formulation has previously considered training a non-trivial latent layer while simultaneously training both the first and second non-linear layers in a feed-forward architecture. New results were required to address the key issues overcome in Sections 3 and 5. The closest precursors that use semidefinite relaxations are supervised latent clustering methods (which were already well reviewed in the paper), but the formulation presented in the paper goes beyond supervised latent clustering.

The experiments were explicitly designed to control the input representations used between the competing learning algorithms. Obviously feature/kernel engineering is important for any practical problem, but the key point of this paper is to demonstrate that non-convexity is not a necessary aspect of successful deep training, despite the prevailing folklore to the contrary. Note that it is possible to use RBF kernels with all the methods presented in this paper, including CVX2. Therefore, it is justified to compare alternative formulations by fixing the input representation to a common shared choice. Changing the input kernel for some methods and not others would only demonstrate the effects of using a poor kernel, but that is an orthogonal issue to the main point of this paper.